# M4V: Multimodal Mamba for Efficient Text-to-Video Generation

## Abstract

Text-to-video generation has significantly enriched content creation and holds the potential to evolve into powerful world simulators. However, modeling the vast spatiotemporal space remains computationally demanding, particularly when employing Transformers, which incur quadratic complexity in sequence processing and thus limit practical applications. Recent advancements in linear-time sequence modeling, particularly the Mamba architecture, offer a more efficient alternative. Nevertheless, its plain design limits its direct applicability to multimodal and spatiotemporal video generation tasks. To address these challenges, we introduce **M4V**, a multimodal Mamba framework for efficient text-to-video generation. Specifically, a MultiModal diffusion Mamba (MM-DiM) block is designed within the framework to enable seamless integration of multimodal information and spatiotemporal modeling. In detail, we introduce a novel multimodal token re-composition design, which employs a bidirectional scheme for multimodal information integration through simple token arrangement, along with visual registers to enhance spatial–temporal consistency. As a result, the MM-DiM blocks in M4V reduce FLOPs by 45% compared with the attention-based alternative when generating videos at $768\times1280$ resolution. Additionally, several training strategies are explored in this work to provide a better understanding of training text-to-video models using only publicly available datasets. Extensive experiments on text-to-video benchmarks demonstrate M4V's ability to produce high-quality videos while significantly lowering computational costs. Code will be made publicly available.

## 1 Introduction

Text-to-video (T2V) generation, which aims at creating video content from natural language instructions, is recognized as one of the most challenging tasks in generative AI. This area has recently received significant attention following the impressive results showcased by OpenAI's Sora (Brooks et al., 2024). Notably, Transformer-based diffusion models, such as DiT (Peebles & Xie, 2023), have been identified as a key factor in achieving Sora's high-quality video synthesis. Despite their potential effectiveness, Transformer-based models suffer from high computational costs due to their quadratic complexity, making the already computationally demanding task even more resource-intensive.

Recently, a novel architecture called Mamba (Gu & Dao, 2023) has demonstrated the potential to match or even surpass Transformers in language modeling tasks. Building on the success of state-space models (SSMs) (Gu et al., 2021), Mamba variants (Dao & Gu, 2024) enhance the long-range modeling capacity of SSMs while maintaining the linear-time complexity in sequence processing. This positions Mamba as a promising alternative to Transformers.

However, unlike Transformers (Vaswani, 2017), which have driven remarkable advancements in generation tasks across both natural language processing and computer vision, Mamba remains largely unexplored in multimodal generative tasks. The limitation arises primarily from the following aspects: (1) Mamba is inherently designed for processing unidirectional 1D sequences, whereas high-resolution image and video generation require sophisticated spatial and temporal modeling capabilities; (2) the lack of design for multimodal interactions, resulting in its limited exploration in text-conditioned visual generation tasks.

In this paper, we address these two limitations by proposing a unified design that leverages Mamba for generating high-fidelity videos from text inputs, results in the MultiModal Diffusion Mamba (MM-DiM) block. Specifically, to model the 3D video distribution, we decouple the information flow into 2D spatial scans and 1D temporal processing, leveraging the autoregressive and unidirectional nature of videos along the temporal dimension. This decoupling enables us to seamlessly exploit the advantages of Mamba without increasing architectural complexity. To address the second limitation of Mamba, we introduce a Multi-Modal Token Re-Composition (MM-Token Re-Composition) strategy before the SSMs. Specifically, to enable multimodal fusion of text and 3D visual information, both text and visual tokens are re-arranged, allowing each modality to perceive global information through hidden states within SSMs.

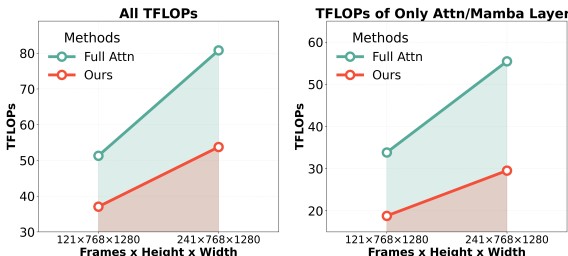

Figure 1: Comparison of FLOPS between full attention baseline and ours.

As a result, our proposed model, named **M4V**, significantly reduces computational FLOPs, particularly in long video generation scenarios, highlighting the advantages of adopting the Mamba structure for this challenging task, as shown in Figure 1. In our experiments, we provide a comprehensive analysis of the model design, demonstrating the potential of Mamba for efficient T2V generation. Additionally, we find that our design generalizes well across base architectures, and also achieving state-of-the-art performance.

Our core contributions are as follows: (1) We introduce M4V, a Mamba-based framework for efficient T2V generation that significantly reduces computational overhead while maintaining the ability to generate high-quality videos. (2) By designing the MultiModal Diffusion Mamba (MM-DiM) block, which enables unified and effective multimodal integration and spatiotemporal modeling, we successfully overcome Mamba's limitations in complex T2V generation. (3) We provide a comprehensive analysis of architectural design choices, demonstrating M4V's efficiency, reasonability and also generalizability across different base architectures.

## 2 RELATED WORK

**Text-to-video Generation.** Text-to-video generation has recently entered a new era, driven by advancements in generative models. The success of diffusion models (Ho et al., 2020) in text-to-image generation (Podell et al., 2023; Esser et al., 2024; Tuo et al., 2023; Guo et al., 2025) has inspired the development of several diffusion-based text-to-video generation models (Ho et al., 2022; He et al., 2022; Chen et al., 2024b). By scaling up Transformer-based diffusion architectures (Peebles & Xie, 2023; Black Forest, 2023), recent models like Sora (Brooks et al., 2024), Kling (Kuaishou, 2024), HunyuanVideo (Kong et al., 2024) have achieved remarkable high-fidelity video generation quality (Wang et al., 2025a; Yin et al., 2024; Xu et al., 2024; Ma et al., 2025; Sand-AI, 2025; Chen et al., 2025; He et al., 2025b). However, the substantial computational costs associated with these approaches severely limit their scalability in both training and deployment. Recently, PyramidFlow (Jin et al., 2024) introduced a novel approach to leverage redundancy in video data by compressing visual tokens both spatially and temporally, achieving significant reductions in training costs. Despite its progress, the quadratic complexity of attention mechanisms continues to constrain the deployment efficiency.

**Mamba and Vision Mamba.** State-space models (SSMs) (Gu et al., 2021) are a family of models inspired by linear-time continuous systems for processing 1D sequences. Despite their efficiency, they are constrained by their time-invariance property, which limits their performance compared with modern large foundation models. To address this limitation, Mamba (Gu & Dao, 2023), a novel form of SSM, introduces time-varying parameters to enhance modeling capacity and employs a hardware-aware selective scan algorithm to maintain linear-time efficiency. The flexibility of Mamba and its variants (Dao & Gu, 2024) enables performance on par with Transformer-based language models. Building on this, recent studies (Waleffe et al., 2024; Lieber et al., 2024; Li et al., 2025b)

show that hybrid architectures combining Mamba and Transformer blocks can achieve strong results in language processing. Motivated by their success, several efforts have extended Mamba to vision tasks. For instance, Zhu et al. (2024); Li et al. (2025a; 2024b); Chen et al. (2024a); Wang et al. (2024a); He et al. (2025a) adapt Mamba for image recognition. Other works (Gao et al., 2024; Hu et al., 2024; Oshima et al., 2024; Fu et al., 2024) explore integrating Mamba with diffusion models for class-conditioned video or image generation, but these studies remain limited to relatively small datasets and do not handle text inputs. In this work, we investigate Mamba for the more challenging task of text-to-video generation, which requires producing high-resolution videos from free-text inputs.

## 3 METHOD

### 3.1 PRELIMINARIES

Before presenting our method, we first provide essential background for clarity. Specifically, we introduce the definition, notation and rationale of the Mamba block. Then, we present an overview of PyramidFlow (Jin et al., 2024), a flow-matching-based (Lipman et al., 2023) video generation model we primarily develop our model based on.

**Mamba.** Originating from the continuous-time linear system

$$h'(\tau) = \mathbf{A}h(\tau) + \mathbf{B}x(\tau), \quad y(\tau) = \mathbf{C}h(\tau) + \mathbf{D}x(\tau), \tag{1}$$

modern state-space models (SSMs) (Gu et al., 2021) process 1-D sequences by discretizing the system with a time-sampling parameter $\Delta$. The continuous parameters $\mathbf{A} \in \mathbb{R}^{n \times n}$ and $\mathbf{B} \in \mathbb{R}^{n \times 1}$ are discretized as

$$\overline{\mathbf{A}} = \exp(\Delta \mathbf{A}), \quad \overline{\mathbf{B}} = (\Delta \mathbf{A})^{-1} \big[ \exp(\Delta \mathbf{A}) - \mathbf{I} \big] \Delta \mathbf{B}, \tag{2}$$

where $n$ is the hidden state size. After discretization, an input sequence $x$ is updated by

$$h^\tau = \overline{\mathbf{A}}h^{\tau-1} + \overline{\mathbf{B}}x^\tau, \quad y^\tau = \mathbf{C}h^\tau + \mathbf{D}x^\tau. \tag{3}$$

Mamba further enhances modeling capacity by making $\mathbf{A}$, $\mathbf{B}$, $\mathbf{C}$, and $\Delta$ input-dependent, coupled with a *selective scan* mechanism for hardware-aware efficiency. This design enables dynamic encoding of global context within the hidden state $h$, allowing information to propagate across the sequence and improving long-range modeling.

**PyramidFlow.** is built on a modified version of FLUX (Black Forest, 2023), a multimodal diffusion transformers, with multi-level latent compressions and an autoregressive prediction paradigm. Specifically, it constructs compressed latent conditions forming "pyramids" along both spatial and temporal dimensions. Along the temporal axis, the prediction of frame $x^i$ is conditioned on the compressed latents of preceding frames:

$$c^i = [K_{\downarrow_2}(x^0), \dots, K_{\downarrow_2}(x^{i-3}), K_{\downarrow_1}(x^{i-2}), x^{i-1}], \tag{4}$$

where $K_{\downarrow_k}(\cdot)$ denotes the compression (downsampling) operation with factor $k$. A temporal causal mask is applied during the attention operation to ensure that earlier frames cannot attend to later ones.

### 3.2 ARCHITECTURE OVERVIEW

Following PyramidFlow, we begin our exploration with the FLUX (Black Forest, 2023) model, and the overall network structure is illustrated in Figure 2(a). Specifically, the FLUX model adopted in PyramidFlow first encodes text and visual information using eight MM-DiT blocks (Esser et al., 2024), with separate parameters for language and visual inputs. Subsequently, sixteen unified Transformer blocks are employed to process text and visual tokens with shared parameters. In our work, we also follow such macro-level architecture and only focus on replacing all subsequent sixteen unified Transformer blocks with our proposed MM-DiM blocks, aiming to explore a **unified** multimodal processing block design based on Mamba. Note that in all of our experiments based on PyramidFlow, we retain the architecture of the eight MM-DiT blocks **unchanged**, as removing their separate parameterization is beyond the scope of this work. In our experiments, we further validate the proposed approach on the recent Wan2.1 model (Wang et al., 2025a) as an extension (see Section 4.3), where our MM-DiM blocks are deployed across all layers.

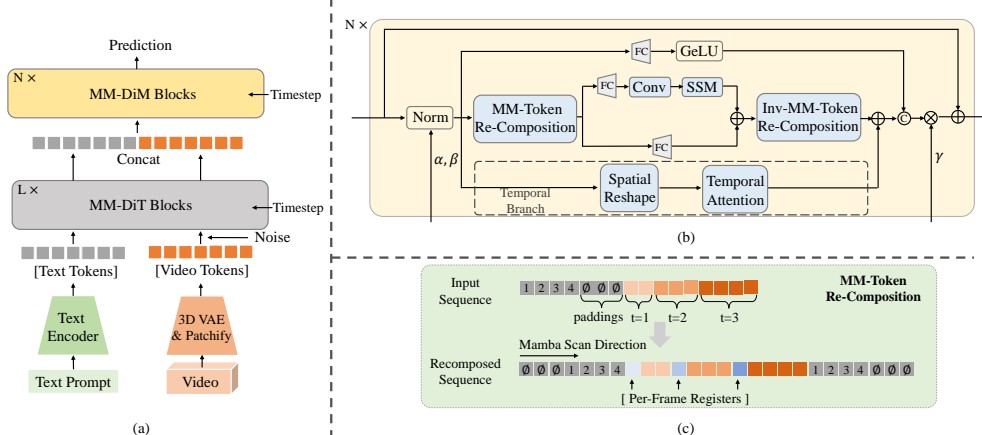

Figure 2: (a) Overview of the generation architecture. (b) Detailed strcture of our MM-DiM Block. $\alpha, \beta, \gamma$ are introduced by projecting the timestep condition, and we ommit the projection for simplicity. (c) Illustration of MM-Token Re-Composition.

## 3.3 MultiModal Diffusion Mamba (MM-DiM) Block

Unlike attention mechanisms that utilize explicit query-key-value (QKV) interactions to integrate context, Mamba faces challenges in handling text conditioning integration. Therefore, prior Mamba-based works (Fei et al., 2024; Wang et al., 2024b; Li et al., 2024b) only process a single modality with Mamba, relying on additional cross-attention for text control. In contrast, our **MM-DiM block** addresses two key challenges for Mamba: (1) facilitating interactions and mutual influence between visual and text tokens, and (2) arranging 3D video latents to seamlessly operate with Mamba. As illustrated in Figure 2(b), an MM-DiM block comprises a major branch that processes multimodal input tokens through an MM-Token Re-Composition operation, followed by an Inv-MM-Token Re-composition step after passing through the SSM. To improve temporal consistency, a light weighted *temporal branch* is incorporated to capture long-range temporal correlations.

**MM-Token Re-Composition.** Different from the attention operation used in PyramidFlow, which requires a causal attention mask to enforce *temporal autoregressive* prediction, state-space models (SSMs) inherently operate in a unidirectional and autoregressive manner but lack the capability for multimodal integration and spatiotemporal awareness. To bridge this gap, as shown in Figure 2(c), we propose the *MM-Token Re-Composition* mechanism, which operates in three key steps:

Text Token Re-Composition. Given the input sequence $X = [Z, X_v]$, where $Z$ denotes the text tokens and $X_v$ the visual tokens, the text tokens $Z$ are first put to the beginning of the sequence, with zero-valued paddings placed on the far left as $Z_l = [\emptyset, Z]$. Starting from a zero-initialized hidden state $h$, this arrangement ensures that $h$ remain zero until the actual text tokens are encountered, as indicated in Equations (1) and (3). Afterward, the visual tokens $X_v$ are appended after the text tokens to enable effective text conditioning, with SSM scaning from left to right. To further facilitate the propagation of visual information back into the text tokens and encourage multimodal alignment, the text tokens are also appended to the end of the sequence with right paddings $Z_r = [Z, \emptyset]$.

Video Token Re-Composition. To mitigate the loss of spatial–temporal information when rearranging a 3D tensor into a 1D sequence, we first adopt the zigzag scanning strategy proposed by Hu et al. (2024) over the spatial dimension. This method alternates between eight distinct scanning paths (see Appendix A.1 for details) across different layers, allowing the global hidden states $h$ in each layer to remain sufficiently diverse to capture rich spatial relationships. Furthermore, as discussed in Section 3.1, the PyramidFlow model dynamically varies the number of conditional frames and their resolutions across pyramid levels. To make the SSM aware of these varying spatial and temporal levels, we introduce *Per-Frame Registers* for the video sequences, inspired by Wang et al. (2025b). Specifically, three types of learnable tokens, corresponding to three different resolution stages, are inserted between conditional frames to (1) signal the start of the next frame and (2) indicate resolution changes. With this design, the visual tokens $X_v = [x^0, \ldots, x^{i-1}, x^i]$ are restructured as $\hat{X}_v = [r^0, x^0, \ldots, r^1, x^{i-1}, r^2]$, where the *Per-Frame Registers* incur negligible computational overhead while significantly enhancing the model's temporal awareness and alignment.

**Inv-MM-Token Re-Composition.** After MM-Token Re-Composition, the input sequence $X$ is transformed into $\hat{X} = [Z_l, \hat{X}_v, Z_r]$ and processed by the SSM to obtain the output $\hat{X}' = [Z_l', \hat{X}_v', Z_r']$. The subsequent Inv-MM-Token Re-Composition operation (1) removes the Per-Frame Registers from the sequence, (2) restores the original visual token order, and (3) aligns and sums the text sequences $Z' = Z_l' + Z_r'$. This operation restores the original sequence structure for processing in the next layer.

**Temporal Branch.** Since pure Mamba models still fall behind Transformers when handling very long contexts (Dao & Gu, 2024; Waleffe et al., 2024), recent studies (Waleffe et al., 2024; Lieber et al., 2024) suggest that the most effective evolution of Mamba is a *hybrid architecture* that combines Transformer and Mamba layers to achieve a better efficiency–performance trade-off.

Different from the heavy block-level hybrid designs in prior works, we propose a lightweight *temporal branch* running in parallel with the main branch, as illustrated in Figure 2(b), to enhance long-range temporal modeling. Specifically, given the conditioning latents $X_C = [x^0, x^1, \ldots, x^{i-1}]$, we first downsample all conditioning frames to the smallest spatial resolution, obtaining $\mathbf{x}_s \in \mathbb{R}^{\frac{H}{K_s} \times \frac{W}{K_s} \times c \times i}$. Next, we flatten the spatial dimensions into the channel dimension to form a short sequence $\mathbf{x}_s \in \mathbb{R}^{i \times S}$, where $S = c \times \frac{H}{K_s} \times \frac{W}{K_s}$. The noisy latent $x^i$ is then partitioned and reshaped into $K_s$ tokens with hidden dimension $S$, and concatenated with the compressed conditioning latents. Finally, a causal attention mechanism is applied along the temporal dimension. The processed latent is reshaped back to its original size and added to the input via a residual connection.

### 3.4 ADDTIONAL IMPROVEMENT

Given the complexity of the T2V task, commonly used public datasets such as WebVid-10M (Bain et al., 2021) are known to be insufficient due to the limited quality of the videos (Zheng et al., 2024). A straightforward strategy adopted by previous works is to scale up the training data (Zheng et al., 2024; Kong et al., 2024; Polyak et al., 2024), in some cases even expanding it to the order of $O(100\text{M})$ samples (Polyak et al., 2024).

Instead of expanding the dataset, we explore a post-training strategy to enhance generation quality. Specifically, suppose the model is trained with the flow-matching objective (Yan et al., 2025). Given the predicted velocity of the last frame $\hat{v}^i$ at a randomly selected timestep $t$ with noise scale $\sigma_t$, where $\sigma_s \leq \sigma_t \leq \sigma_e$ ($\sigma_s = 0$, $\sigma_e = 1$ in Lipman et al. (2023)), we assume the predicted velocity approximates the true velocity, i.e., $\hat{v}^i \approx v^i = x_e^i - x_s^i$. Using this, a one-step denoising operation is performed to obtain

$$\hat{x}_1^i = \frac{1}{\sigma_e}\left(x_t^i + \frac{\sigma_e - \sigma_t}{\sigma_e - \sigma_s}\hat{v}^i - (1 - \sigma_e)x_0^i\right), \tag{5}$$

where $x_0^i$ is the noisy input and $\hat{x}_1^i$ is the predicted clean latent. The latent $\hat{x}_1^i$ is then decoded and evaluated using reward models $r_1$ and $r_2$, where we adopt HPSv2 (Wu et al., 2023) and CLIP (Radford et al., 2021) in our experiments:

$$\mathcal{L}_{\text{reward}} = -r_1(D(\hat{x}_1^i)) - r_2(D(\hat{x}_1^i)), \tag{6}$$

where $D$ denotes the decoder of the 3D VAE. This reward loss is backpropagated as additional supervision to refine the generation of each frame.

## 4 EXPERIMENTS

### 4.1 EXPERIMENTAL SETTINGS

**Training Dataset.** Our model is trained on a diverse and extensive collection of publicly available and proprietary image and public video datasets. For image data, we utilize the LAION-aesthetic dataset (Schuhmann et al., 2022), 40 million synthetic images generated by Midjourney, 40 million images sourced from Instagram, and 10 million internally curated portrait images. For video data, we incorporate WebVid-10M (Bain et al., 2021), OpenVid-1M (Nan et al., 2024), 1 million high-resolution, watermark-free videos from the Open-Sora Plan (PKU-Yuan Lab, 2024). After preprocessing, the final training set comprises approximately 10 million single-shot video clips.

**Implementation Details.** Our primary study is built upon the PyramidFlow framework, which enables efficient training with both spatial and temporal pyramids. Unless otherwise specified,

Table 1: Benchmark results on VBench (Huang et al., 2024). The best results among models trained on public data are marked in **bold**. †: Reproduced results using official code and the same training data as in our experiments. *: Models that are initialized (or partially initialized) from public models.

| Model | Video Training Data | Total Score | Quality Score | Semantic Score | Motion Smoothness | Dynamic Degree | Aesthetic Quality | Imaging Quality |
|---|---|---|---|---|---|---|---|---|
| Gen-2 | Proprietary | 80.58 | 82.47 | 73.03 | **99.58** | 18.89 | 66.96 | 67.42 |
| CogVideoX-5B | Proprietary | 81.61 | 82.75 | 77.04 | 96.92 | 70.97 | 61.98 | 62.90 |
| Kling | Proprietary | 81.85 | 83.38 | 75.68 | 99.40 | 46.94 | 61.21 | 65.62 |
| Gen-3 Alpha | Proprietary | 82.32 | 84.11 | 75.17 | 99.23 | 60.14 | 63.34 | 66.82 |
| HunyuanVideo | Proprietary | 83.24 | 85.09 | 75.82 | 98.99 | 70.83 | 60.36 | 67.56 |
| Wan2.1 | Proprietary | 84.70 | 85.64 | 80.95 | 96.92 | 94.35 | 61.53 | 67.28 |
| VideoCrafter2* | Public | 80.44 | 82.20 | 73.42 | 97.73 | 42.50 | 63.13 | 67.22 |
| T2V-Turbo* | Public | 81.01 | 82.57 | 74.76 | 97.34 | 49.17 | 63.04 | **72.49** |
| Open-Sora Plan v1.1 | Public | 78.00 | 80.91 | 66.38 | 98.28 | 47.72 | 56.85 | 62.28 |
| Open-Sora 1.2 | Public | 79.76 | 81.35 | 73.39 | 98.50 | 42.39 | 56.85 | 63.34 |
| Pyramidflow | Public | 81.72 | 84.74 | 69.62 | 99.12 | 64.63 | 63.26 | 65.01 |
| Pyramidflow† | Public | 81.61 | 83.54 | 73.90 | 99.32 | 66.66 | 63.96 | 61.69 |
| M4V (Pyramidflow) | Public | 81.55 | 83.31 | 74.47 | 99.33 | 60.55 | 64.08 | 62.22 |
| M4V* (Wan2.1) | Public | **86.14** | **87.56** | **80.45** | 99.18 | **96.70** | **67.52** | 65.62 |

all results reported in the following experiments are based on the PyramidFlow framework, with the scope of replacing all unified Transformer blocks with our MM-DiM blocks. To accelerate the training of Mamba blocks, inspired by Wang et al. (2024c), we initialize part of the parameters in Mamba with pre-trained attention weights (see Section D.2 for more details). Besides, we introduce linearly increasing levels of corruptive noise to the conditioning frames, which improves the training stability in the early training stages. To efficiently pretrain the M4V model, we adopt a progressive training strategy. The process begins with text-to-image (T2I) training at 384p resolution, then gradually increase from 384p to 768p and extend the video length from 57, 121 and 241 frames, training with both image and video data. This staged approach facilitates stable adaptation to more complex tasks and longer token sequences. More details can be found in Sections A.3 and D.

Besides, to validate the generalizability of our design, we further extend our method to the recent Wan2.1 (Wang et al., 2025a) framework. For this extension, we directly replace all self-attention layers in Wan2.1 with our MM-DiM blocks, and partially initialize the network from the official Wan2.1 pre-trained weights following a strategy similar to Wang et al. (2024c). To facilitate comparison with the official Wan2.1, we adopt the same resolutions and frame numbers as in Table 2. We reuse Wan2.1's text encoder and VAE. As Wan2.1's framework is non-autoregressive and does not employ a pyramid structure, M4V (Wan2.1) and M4V (PyramidFlow) differ in data formatting and loss computation during training. M4V (Wan2.1) does not perform pyramidal downsampling of video latents, and both flow matching loss calculation and reward learning are applied to the entire video latent; thus, inference is also performed on the whole video latent simultaneously. Aside from these differences, other training settings, such as the training data and stages, are consistent with our M4V (PyramidFlow). Due to the increased memory requirements of M4V (Wan2.1), we utilize DeepSpeed Zero Stage 3 for optimized training. The 480p T2V/T2I hybrid training is conducted using a learning rate of $1 \times 10^{-4}$, 128 GPUs, and 60k steps. For 720p, we train the model with a learning rate of $5 \times 10^{-5}$, using 128 GPUs for 35k steps.

**Evaluation Metrics.** For quantitative comparison, we utilize VBench (Huang et al., 2024), a widely adopted benchmark designed for comprehensive T2V evaluation. VBench assesses T2V performance using 1,000 prompts that cover diverse scenarios. For each text prompt, we generate five videos using different random seeds, each containing 121 frames at 768p resolution for evaluation. The *Total Score* on VBench is computed as a weighted average of the *Quality Score* and *Semantic Score*. For more details, please refer to Section B.1 and the VBench paper.

**Fast Evaluation Protocol.** For architectural-level design studies, conducting full-cycle training for each variant during ablation is computationally prohibitive, often requiring thousands of GPU hours, and VBench evaluation is also time-intensive. To address this, we propose a fast evaluation protocol. Specifically, in our ablation studies, each Mamba variant is trained at 5s–384p for only 20k steps, starting from the same initialization of an attention-based model pre-trained on 2-second videos. Generation performance is evaluated using 50 prompts sampled from VBench. Since not all metrics can be reliably computed on this subset, we report only a subset of evaluation metrics that best capture the differences between variants, including *subject consistency*, *aesthetic quality*, *image quality*, and *overall consistency*. Although this fast evaluation protocol may not perfectly reflect the



Figure 3: User study between Ours, T2V-Turbo, CogvideoX, HunyanVideo and Pyramidflow.

behavior of fully trained models, it provides *relative* performance trends at early training stages that effectively guide architectural design. For further details, please refer to Appendix C.

**Human Evaluation**. In addition to automated metrics, we also include a user study to assess human preferences of generated videos from different models. Following the setup in Jin et al. (2024), we selected 50 video prompts sourced from both VBench and the Internet. Participants were asked to rank videos based on three criteria: aesthetic quality, motion smoothness, and semantic coherence.

## 4.2 EFFICIENCY ANALYSIS AND COMPARISON

The integration of Mamba significantly reduces computational overhead in video generation compared to recent models that heavily rely on 3D full-attention structures. For a video with $T$ frames and $M$ spatial tokens, the complexity of full-sequence attention, as used in DiT, is $\mathcal{O}((TM)^2)$. In contrast, the SSM requires only $\mathcal{O}(TM)$, and the temporal attention adds $\mathcal{O}(T^2)$. Given that $T \ll M$, the MM-DiM block achieves a substantial improvement in training efficiency with an overall complexity of $\mathcal{O}(TM + T^2)$. We provide quantitative analysis in terms of TFLOPs and shown in Table 4. Notably, for generating a 241-frame video, our model reduces the computational cost of the mixer layers by 45% (from 55.44 to 29.52 TFLOPs).

Table 2: Generation speed comparison across models.

| Model | Video Size | Time (s)↓ |
|---|---|---|
| HunyuanVideo | $720 \times 1280 \times 129$ | 1890 |
| PyramidFlow | $768 \times 1280 \times 241$ | 812 |
| M4V (PyramidFlow) | $768 \times 1280 \times 241$ | **613** |
| Wan2.1 | $720 \times 1280 \times 81$ | 1700 |
| M4V (Wan2.1) | $720 \times 1280 \times 81$ | **1210** |

**Efficiency Comparison.** In Table 2, our M4V achieves substantially faster generation at high resolutions and long video sequences. Noted that different models are trained with varying video resolutions and frame counts, making a strictly fair efficiency comparison across all methods infeasible.

## 4.3 MAIN RESULTS

**Quantitative Results.** We evaluate M4V's text-to-video generation performance and compare it with other methods on VBench (Huang et al., 2024), as shown in Table 1. First, with PyramidFlow as baseline method, M4V achieves a comparable Total Score to PyramidFlow (81.55% vs. 81.61%), while significantly reducing computational cost, as discussed in Section 4.2. Besides, when extending our method to the recent Wan2.1 model. By replacing all self-attention layers with our proposed MM-DiM blocks and fine-tuning on our training data, we surprisingly observe both improved performance in Table 1 over the original Wan2.1 and increased inference efficiency in Table 2.

**Human Preference**. To better understand how our method compares to other approaches based on human judgments, we conducted a user study comparing our model with four state-of-the-art baselines: PyramidFlow, CogVideoX (Yang et al., 2024), T2V-Turbo (Li et al., 2024a), and HunyuanVideo (Kong et al., 2024). As illustrated in Figure 3, our method shows a clear advantage in semantic coherence and motion smoothness compared to open-source models such as PyramidFlow, CogVideoX, and T2V-Turbo. Although our model lags behind HunyuanVideo in these two aspects, it outperforms it in aesthetic quality. This result is consistent with the quantitative evaluation of *Aesthetic Quality* on VBench, where HunyuanVideo tends to produce overly realistic videos. A detailed description of the user study setup, selected prompts, and comprehensive results can be found in Section B.1, and corresponding video samples are included in the supplementary materials.

## 4.4 EFFECT OF MODEL DESIGN

This section provides a comprehensive ablation study to evaluate the effectiveness of MM-DiM block based on our *Fast Evaluation Protocol* described in Section 4.1.

Table 3: Ablation study of the model architecture using the proposed fast evaluation protocol. **Text**: Enables bi-directional information aggregation through text token re-composition. **Vis**: Adds per-frame registers within the visual sequence. Temp: Incorporates a temporal branch within each block. **Overall-Con** measures the consistency between the generated video and the input text, while the other metrics assess different aspects of video quality. Significant metric changes with **Text** and **Vis** are highlighted for clarity.

| Text | Vis | Temp | Sub-Cons | Aes-Qual | Img-Qual | Overall-Cons | Avg. |
|------|-----|------|----------|----------|----------|--------------|------|
|      |     |      | 93.28 | 46.60 | 63.16 | 19.77 | 55.70 |
| ✓    |     |      | 92.19 | 45.39 | 54.83 | 21.23 | 53.41 |
|      | ✓   |      | 95.41 | 48.69 | 64.18 | 18.86 | 56.79 |
| ✓    | ✓   |      | 93.53 | 49.82 | 63.79 | 21.26 | 57.10 |
| ✓    | ✓   | ✓    | **95.67** | **51.25** | **66.38** | **21.68** | **58.75** |

Table 4: Computational analysis of architectural designs. TFLOPs are calculated for mixer layers, *i.e.*, attention or Mamba. Both TFLOPs and inference time are estimated at 768p resolution, on a single NVIDIA A100 GPU.

| Model | Params (B) | TFLOPs | | Inference Time (s) | | Avg. Score |
|-------|-----------|--------|--|--------------------|--|------------|
|       |           | 121-frms | 241-frms | 121-frms | 241-frms | |
| Full Attn | 1.97 | 33.84 | 55.44 | 296 | 812 | 59.84 |
| *Parallel* | 2.21 | 50.19 | 82.03 | 313 | 858 | 59.97 |
| *Post-half* | 2.00 | 25.20 | 41.04 | 224 | 661 | 58.17 |
| *Pre-half* | 2.00 | 25.20 | 41.04 | 224 | 661 | 58.69 |
| *Interleaved* | 2.00 | 25.20 | 41.04 | 224 | 661 | 58.60 |
| *Full* | 2.04 | 16.35 | 26.64 | 210 | 570 | 57.10 |
| *Full+Temp-Branch* | 2.21 | 18.80 | 29.52 | 226 | 613 | 58.75 |

**Component-wise Ablation.** We systematically ablate each component of our block design, as summarized in Table 3. The first row excludes our proposed designs, using only zigzag scan paths (Hu et al., 2024) along the spatial dimension and adding positional encodings to each token for processing with Mamba. First, incorporating Text Token Re-Composition significantly improves the *Overall Consistency* metric, indicating enhanced text–video alignment, although the text-focused design may slightly degrade certain visual quality metrics. Next, integrating Per-Frame Registers into the video sequence improves all video quality metrics, demonstrating their effectiveness in helping Mamba capture spatial–temporal dependencies. When combined, these two components lead to consistent gains across all metrics compared to the baseline. Finally, adding the lightweight temporal branch further enhances performance on all reported metrics, reflecting the complementary benefits of combining SSMs with attention mechanisms.

**Design of Model Structure.** In light of recent hybrid architectures (Waleffe et al., 2024; Lieber et al., 2024), we investigate how different configurations of attention and Mamba blocks influence both computational cost and generation quality. Specifically, based on the overall architecture described in Section 3.2, we ablate the block choice for the sixteen target blocks using the following configurations: (1) *Full*: all target blocks employ Mamba; (2) *Post-half*: Mamba is applied only to the latter half of the target blocks, with attention used in the first half; (3) *Pre-half*: Mamba is applied to the first half of the target blocks, with attention used in the latter half; (4) *Parallel*: Mamba and attention operate in parallel within each target block, sharing the same MLP layers; (5) *Interleaved*: Mamba and attention alternate sequentially across the target blocks.

Variant *Full* is expected to yield the lowest computational cost but may suffer from limited modeling capacity. In contrast, *Parallel* leverages the complementary strengths of Mamba and attention across all layers, but at the highest computational cost.

To ensure a fair comparison, all variants are trained with same steps and using the fast evaluation protocol as described in Section 4.1. The performance results are presented in Table 4. Among all variants, *Parallel* achieves the highest performance but comes with a substantial computational cost while offering only a marginal 0.09% improvement over full attention. The *Full* variant, which applies Mamba to all blocks, significantly reduces computational overhead with comparable performance with others. When incorporating the proposed temporal branch to *Full*, the model achieves the best overall scores while maintaining lower computational costs.

## 4.5 EFFECT OF ADDITIONAL TRAINING DESIGNS

As discussed in Section 3.4, we additionally explore training strategies to improve generation quality given the limitation of public datasets.

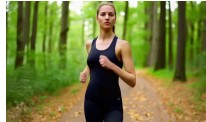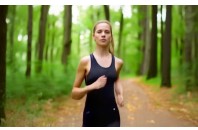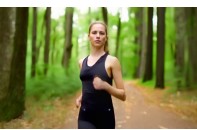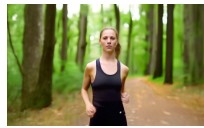

(a) A determined individual in a sleek, black athletic outfit jogs along a winding forest trail, surrounded by towering trees and dappled sunlight filtering through the leaves.

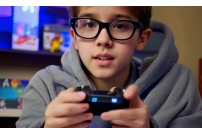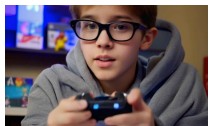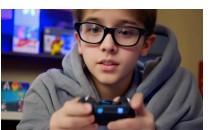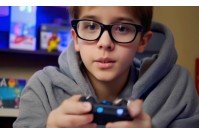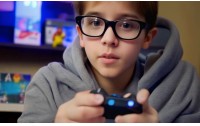

(b) A young person, wearing a cozy gray hoodie and black-rimmed glasses, sits in a dimly lit room, intensely focused on a video game. The glow from the TV screen illuminates their face.

Figure 4: Visualization of text-to-video generation results which are generated at 5s, 768p, 24fps.

First, we consider using reward models for post-training, named *Reward Learning*, and the results are shown in Table 5. With an additional post-training stage using $\mathcal{L}_{\text{reward}}$, the generation performance improves by 0.16% on VBench. We also provide a visual analysis about our $\mathcal{L}_{\text{reward}}$ in Figure 5.

Table 5: Ablation study of training improvements on official VBench (Huang et al., 2024).

| Training Design | | Total Score | Quality Score | Semantic Score |
|---|---|---|---|---|
| Reward Learning | Generated Data | | | |
| | | 81.55 | 83.31 | 74.47 |
| ✓ | | 81.71 | 83.32 | 75.27 |
| | ✓ | 81.59 | 83.35 | 74.52 |
| ✓ | ✓ | **81.91** | **83.36** | **76.10** |

As shown, such post-training can visually help the model to correct undesirable motions, resulting in outputs that more closely align with the input prompts, thereby enhancing prompt adherence and improving the Semantic Score in Table 5.

Furthermore, we conduct an initial investigation on augmenting our training data with approximately 80,000 videos synthesized using HunyuanVideo (Kong et al., 2024) with prompts provided by GPT-4o, primarily depicting subjects engaged in diverse motion activities. Incorporating these generated videos into the final-stage training further enhances generation performance, particularly when combined with our $\mathcal{L}_{\text{reward}}$.

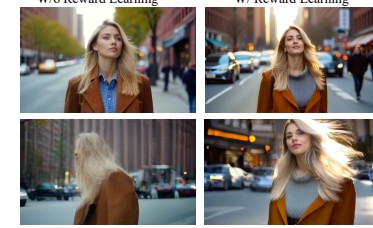

Figure 5: Visual analysis of reward learning.

### 4.6 VISUAL RESULTS

Figure 4 shows some visual results generated by our model. With our advanced design, the model is able to produce visually consistent videos with high aesthetic quality. Additional T2V and image-conditioned T2V results can be found in Section E and the supplementary materials.

## 5 CONCLUSION AND LIMITATIONS

In this work, we introduce **M4V**, a Mamba-based framework for text-to-video generation. Given the multimodal nature of this task, we propose the **MultiModal Diffusion Mamba (MM-DiM) Block**, a unified module that overcomes Mamba's inherent limitations in multimodal modeling. As a result, M4V achieves substantial reductions in computational cost while maintaining high generation quality. Our experiments provide a comprehensive analysis of architectural design choices with respect to both performance and efficiency, demonstrating the potential of linear-time models as a compelling alternative to attention-based methods. Furthermore, extending our design to the recent Wan2.1 model confirms the generalizability of our approach. We also explore additional training strategies that further enhance generation quality when using only publicly available datasets.

**Limitations.**. While our lightweight temporal branch complements the MM-DiM block effectively, it potentially can be replaced by a Mamba design, though the efficiency gain may be marginal. Moreover, when using PyramidFlow as the base model, M4V exhibits slightly lower performance on the Dynamic Degree metric, possibly due to the multi-level compression in PyramidFlow, which makes it more challenging for the SSM hidden states to capture global information. Future work would include further exploring improved temporal modeling within the Mamba framework.

ETHICS STATEMENT

This work does **not** include any human subjects, potentially harmful insights, privacy, sensitive, or personal information that may raise questions regarding the Code of Ethics.

REPRODUCIBILITY STATEMENT

Reproducibility is a key goal in our development for contributing to the text-to-video generation community. In this work, all experiments are conducted using publicly available datasets. The main paper and appendix provide complete details on data preprocessing, model architectures, training procedures, and evaluation protocols. We will make our models and code, including both training and evaluation scripts, publicly available upon acceptance.

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

## A    IMPLEMENTATION DETAILS

### A.1    SPATIAL SCAN PATH IN MM-DIM BLOCK.

As shown in Figure 6, we follow (Hu et al., 2024) to apply eight type of scan paths along the spatial dimension for Mamba, which include:

(a) top-left to the bottom-right, following a "downward first, then rightward" direction.

(b) top-left to the bottom-right, following a "downward right, then downward" direction.

(c) bottom-left to the top-right, following a "upward first, then rightward" direction.

(d) bottom-left to the top-right, following a "rightward first, then upward" direction.

(e) bottom-right to the top-left, following a "upward first, then leftward" direction.

(f) bottom-right to the top-left, following a "leftward first, then upward" direction.

(e) top-right to the bottom-left, following a "downward first, then leftward" direction.

(f) top-right to the bottom-left, following a "leftward first, then downward" direction.

Following (Hu et al., 2024), we apply a single type of scan path per layer, while alternating the type across layers.

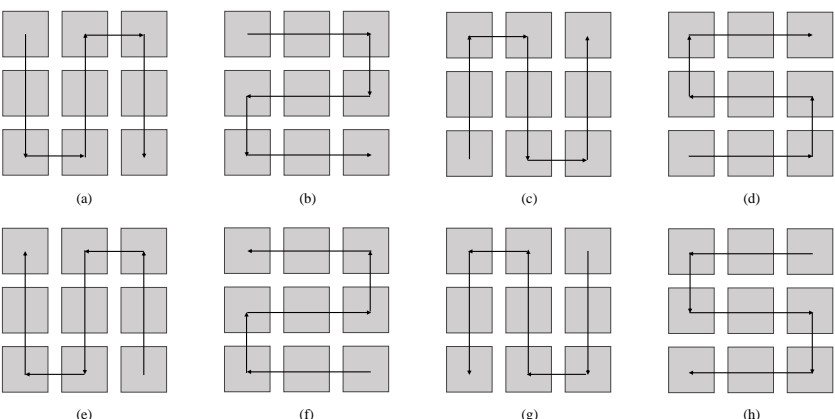

Figure 6: Sptial scan paths for Mamba.

### A.2    OVERALL ARCHITECTURE

We retain $L = 8$ *dual-stream blocks* from MM-DiT (Esser et al., 2024), which use separate parameter sets for different modalities. While replacing all dual-stream modules with Mamba blocks could theoretically reduce TFLOPs further (from 29.52 to 22.416), our preliminary experiments reveal that this substitution introduces significant latency during training (approximately a $1.5\times$ increase per iteration). This overhead arises from the need to separately process text and video tokens for Mamba, involving additional reshaping, slicing, concatenation, and causal `conv1d` operations to prepare inputs for the state-space model (SSM). As a result, full replacement proves impractical for our architecture-level exploration.

Therefore, our design modifications focus exclusively on the *single-stream* blocks, where the M4V model employs MM-DiM blocks throughout. This choice ensures a balance between efficiency gains and manageable training costs. In future work, this limitation could potentially be mitigated through engineering optimizations, for instance, by developing custom PyTorch kernels to better utilize GPU resources. While such improvements would enhance runtime efficiency, they may also reduce flexibility for iterative architectural evolvement.

### A.3    LINEAR SCALING TEMPORAL CORRUPTION NOISE

As introduced in Section 3.1, the prediction of a frame $x^i$ is conditioned on

$$c^i = [K_{\downarrow_2}(x^0), \ldots, K_{\downarrow_2}(x^{i-3}), K_{\downarrow_1}(x^{i-2}), x^{i-1}]. \tag{7}$$

However, due to the error accumulation during autoregressive video generation, latter frames tend to have lower quality than previous ones. Besides, as indicated by (Valevski et al., 2024), using clean latent during training would lead to training-inference inconsistency. Therefore, similar to PyramidFlow (Jin et al., 2024), we add corruption noise to the condition frames during training. However, different from PyramidFlow, we use a linear scaling corruption noise instead of a constent noise across frames. Specifically, given a corruption ratio $\eta$, we randomly sample the maximum corruption scale $\eta_t^{max}$ in range $[0, \eta]$ and a minimum corruption scale $\eta_t^{min}$ in range $[0, \eta_t^{max}]$. Then we add per-frame noise to the condition frames with noises $[\sigma_{\eta_t^{min}}, \ldots, \sigma_{\eta_t^{max}}]$ with linear intervals. This design brings slight improvement in convergence speed in our early training stages.

## B  EXPERIMENTAL SETTINGS.

### B.1  QUANTITATIVE EVALUATION SETTING

In this work, we include various baseline methods for comparisons on VBench. Specifically, we include fully open-sourced methods including Open-Sora Plan (PKU-Yuan Lab, 2024), Open-Sora 1.2 (Zheng et al., 2024), and PyramidFlow (Jin et al., 2024) for our major comparison. We also include approaches that uses proprietary data for reference, including Pika 1.0 (art, 2024), CogVideoX (Yang et al., 2024), Kling (Kuaishou, 2024), Runway Gen-3 Alpha (Runway, 2024), and HunyuanVideo (Kong et al., 2024). VideoCrafter2 (Chen et al., 2024b), T2V-Turbo (Li et al., 2024a), Vchitect-2.0 (Fan et al., 2025). We directly source the results for all methods from the official leaderboard for comparison. All compared baseline methods are based on attention for spatialtemporal modeling, while we use Mamba instead.

VBench is an automatic benchmark designed for text-to-video generation models. It scores each submission along *sixteen* objective dimensions that jointly capture (i) low-level visual fidelity—e.g. absence of flicker, smooth motion and high aesthetic / imaging quality—and (ii) high-level semantic faithfulness such as correct object classes, actions, colours and scene composition. For every prompt the model must generate five clips; the per-metric scores are averaged, then linearly normalised with official *min–max* statistics and multiplied by a dimension weight (dynamic-degree is down-weighted to 0.5, all others to 1.0). The normalised scores are grouped into a Quality Score (7 metrics) and a Semantic Score (9 metrics). As summarised in Table 6, VBench reports the weighted mean of each block and finally fuses them with a 4:1 ratio so that perceptual quality carries four times the importance of semantic accuracy:

$$\text{Total Score} = \frac{4\,\text{Quality} + 1\,\text{Semantic}}{5}.$$

This single scalar is used for leaderboard ranking, while the two component scores still expose a model's individual strengths and weaknesses.

Table 6: Composition of the VBench headline scores.

| Score | Included sub-metrics |
|---|---|
| Quality Score | subject consistency; background consistency; temporal flickering; motion smoothness; aesthetic quality; imaging quality; dynamic degree (0.5× weight) |
| Semantic Score | object class; multiple objects; human action; color; spatial relationship; scene; appearance style; temporal style; overall consistency |
| Total Score | $\text{Total} = \dfrac{4 \times \text{Quality} + 1 \times \text{Semantic}}{5}$ |

### B.2  HUMAN PREFERENCE SETTING

In order to evaluate the performance of our method, we conducted a user study to compare it against four state-of-the-art (SOTA) models: PyramidFlow, CogVideoX, T2V-Turbo, and HunyuanVideo.

The user study aimed to assess the generated video quality across three key aspects: aesthetic quality, motion smoothness, and semantic coherence. The design and methodology of the study are outlined as follows.

The study involved five methods: our proposed approach and the four SOTA models mentioned above. A total of 50 video prompts were selected for evaluation, sourced from both VBench and the Internet. These prompts were carefully chosen to cover a broad spectrum of content types and video scenarios, ensuring that the evaluation reflects a diverse range of real-world use cases.

Over 50 participants, including both experts and non-experts, took part in the study. Each participant was asked to rank the generated videos for each method based on three criteria: aesthetic quality, motion smoothness, and semantic coherence. The ranking scale used was from 1 to 5, with 1 representing the highest quality and 5 representing the lowest.

For each of the 50 prompts, the participants were shown videos generated by all five methods, and they were asked to assign a score to each model in the three evaluation categories. The win rates between ours and the compared methods were then aggregated across all participants.

### B.3 DETAIL RESULTS OF USER STUDY

We list the average ranking of all prompts in Fig. 7 and all prompts bellow:

**1.** A breathtaking coastal beach in spring, where gentle waves caress the golden sand in super slow motion. The scene captures the delicate dance of turquoise waters, each wave rolling gracefully and retreating with a soft whisper.

**2.** A bustling city street comes alive with vibrant energy, lined with towering skyscrapers and historic buildings. The scene captures the essence of urban life, with people of all ages and backgrounds walking briskly, some carrying shopping bags, others engaged in animated conversations.

**3.** A bustling hospital corridor, filled with the soft hum of activity, features doctors in white coats and nurses in scrubs moving purposefully. The walls are adorned with calming artwork and information.

**4.** A bustling train station platform comes to life in the early morning light, with commuters clad in winter coats and scarves, their breath visible in the crisp air. The platform is lined with vintage lampposts casting a warm glow, and a sleek, modern train pulls in, its doors sliding open with a soft hiss.

**5.** A charming panda, wearing a chef's hat and a red apron, stands in a cozy, rustic kitchen filled with wooden cabinets and colorful utensils. The panda carefully chops vegetables on a wooden cutting board, its furry paws moving with surprising dexterity.

**6.** A cheerful individual stands in a lush backyard, surrounded by vibrant greenery and blooming flowers, tending to a sizzling barbecue grill. They wear a red apron over a casual white t-shirt and jeans, with a chef's hat perched jauntily on their head.

**7.** A colossal, hyper-realistic spaceship descends gracefully onto the rugged Martian surface, its sleek metallic hull reflecting the crimson hues of the planet. Dust and small rocks scatter as the landing thrusters engage, creating a dramatic cloud of Martian soil.

**8.** A contemplative individual, dressed in a dark, hooded jacket, stands alone on a dimly lit urban street, the soft glow of streetlights casting long shadows. They lift a cigarette to their lips, the ember glowing brightly in the night.

**9.** A cozy, dimly-lit restaurant exudes warmth and charm, with rustic wooden tables adorned with flickering candles and fresh flowers. Soft, ambient music plays in the background, enhancing the serene atmosphere.

**10.** A cozy, dimly-lit restaurant with rustic wooden tables and chairs, adorned with flickering candles and fresh flowers in glass vases, creates an intimate ambiance. The walls are lined with vintage photographs and shelves filled with wine bottles, adding a touch of nostalgia.

**11.** A determined individual in a sleek, black athletic outfit jogs along a winding forest trail, surrounded by towering trees and dappled sunlight filtering through the leaves. Their rhythmic

strides create a sense of purpose and focus, with the soft crunch of leaves underfoot adding to the serene ambiance.

**12.** A determined individual, dressed in a red flannel shirt, blue jeans, and sturdy boots, pushes a weathered wooden cart along a narrow, cobblestone street. The scene is set in a quaint, old-world village with charming stone buildings and ivy-covered walls.

**13.** A drone captures a breathtaking aerial view of a festive celebration in a snow-covered town square, centered around a towering, brilliantly lit Christmas tree adorned with twinkling lights and ornaments.

**14.** A fluffy orange cat with striking green eyes sits calmly to the right of a large, friendly golden retriever, both facing the camera. The cat's fur is meticulously groomed, and it wears a small, elegant collar with a bell.

**15.** A golden retriever with a shiny coat strolls leisurely through a sun-dappled forest path, the morning light filtering through the trees casting a warm glow. The dog's tail wags gently as it sniffs the air, ears perked up, taking in the serene surroundings.

**16.** A grand, historic mansion stands majestically atop a hill, its stone facade adorned with ivy and intricate carvings, bathed in the golden light of a setting sun. The camera pans to reveal tall, arched windows reflecting the vibrant hues of the sky, while the meticulously manicured gardens, with their blooming flowers and ornate fountains, add a touch of elegance.

**17.** A joyful dog, a golden retriever, sits proudly in a vibrant yellow turtleneck, its fur contrasting beautifully against the dark studio background. The dog's eyes sparkle with happiness, and its mouth is open in a cheerful pant, showcasing its playful nature.

**18.** A joyful individual, bundled in a red winter coat, knitted hat, and gloves, stands in a snow-covered park, rolling a large snowball to form the base of a snowman. The scene is set against a backdrop of snow-laden trees and a serene, overcast sky.

**19.** A joyful, fuzzy panda sits cross-legged by a crackling campfire, strumming a small acoustic guitar with enthusiasm. The panda's black and white fur contrasts beautifully with the warm glow of the fire.

**20.** A lone adventurer, clad in a bright red life jacket and a wide-brimmed hat, paddles a sleek, yellow kayak through a serene, crystal-clear lake surrounded by towering pine trees and majestic mountains.

**21.** A lone astronaut, clad in a pristine white spacesuit with reflective visors, floats gracefully against the vast, star-studded expanse of space. As the camera pans left, the astronaut's movements are slow and deliberate, capturing the serene beauty of weightlessness.

**22.** A lone rider, clad in a sleek black leather jacket, matching helmet, and dark jeans, navigates a winding mountain road on a powerful motorcycle. The sun sets behind the peaks, casting a golden glow on the rugged landscape.

**23.** A lone stormtrooper, clad in iconic white armor, stands on a sunlit beach, holding a futuristic vacuum cleaner. The scene opens with the stormtrooper methodically vacuuming the golden sand, the ocean waves gently lapping in the background.

**24.** A majestic steam train, with its vintage black and red carriages, chugs along a winding mountainside track, enveloped in a cloud of white steam. The train's powerful engine, adorned with brass accents.

**25.** A playful panda, with its distinctive black and white fur, sits on a wooden swing set in a lush bamboo forest. The panda's eyes sparkle with joy as it grips the ropes tightly, swaying back and forth.

**26.** A playful squirrel, with its bushy tail flicking, sits on a park bench, holding a miniature burger in its tiny paws. The scene is set in a vibrant, sunlit park with lush green grass and colorful flowers in the background.

**27.** A plump rabbit, adorned in a flowing purple robe with golden embroidery, ambles through an enchanting fantasy landscape. The rabbit's large, expressive eyes take in the vibrant surroundings, where towering mushrooms with glowing caps and bioluminescent flowers light up the path.

**28.** A plush teddy bear, with soft brown fur and a red bow tie, stands on a stool in a cozy, vintage kitchen. The bear's tiny paws are submerged in a sink filled with soapy water, bubbles floating around.

**29.** A pristine white bicycle stands alone on a cobblestone street, its sleek frame and vintage design catching the morning light. The bike is adorned with a wicker basket on the front, filled with fresh flowers, adding a touch of charm.

**30.** A pristine white cat with striking blue eyes lounges gracefully on a sunlit windowsill, its fur glistening in the warm afternoon light. The cat stretches luxuriously, its paws extending and tail curling elegantly.

**31.** A quaint bakery shop, bathed in warm, golden light, showcases an inviting display of freshly baked goods. The rustic wooden shelves are lined with an assortment of crusty baguettes, flaky croissants, and golden-brown pastries, each meticulously arranged.

**32.** A refined couple, dressed in elegant evening attire, navigates a bustling street under a heavy downpour. The man, in a tailored black tuxedo, and the woman, in a flowing crimson gown, both hold delicate paper umbrellas adorned with intricate patterns.

**33.** A serene cow with a glossy brown coat lies comfortably on a bed of fresh straw inside a rustic, sunlit barn. The gentle rays of the afternoon sun filter through the wooden slats, casting a warm, golden glow over the scene.

**34.** A serene individual sits in a cozy, sunlit nook, surrounded by shelves filled with books, wearing a soft, oversized sweater and glasses. They hold an old, leather-bound book, its pages slightly yellow.

**35.** A serene individual, dressed in a flowing white blouse and light blue jeans, stands at a rustic wooden table in a sunlit room filled with greenery. They carefully select vibrant blooms from a wicker basket, including roses, lilies, and daisies, and begin arranging them in a crystal vase.

**36.** A skilled artisan, wearing protective gloves and a welding mask, stands in a dimly lit workshop filled with tools and metal scraps. The person carefully heats a metal rod with a blowtorch, the orange flames casting a warm glow on their focused face.

**37.** A sleek Mars rover, equipped with advanced scientific instruments and cameras, traverses the rugged, reddish terrain of the Martian surface. The scene opens with a panoramic view of the barren landscape, featuring rocky outcrops and distant mountains under a dusty, pinkish sky.

**38.** A sleek, black motorcycle with chrome accents roars to life on an open highway, its rider clad in a black leather jacket, helmet, and gloves. The camera captures a close-up of the rider's gloved hand.

**39.** A sleek, modern train glides effortlessly along the tracks, its metallic exterior gleaming under the bright midday sun. The train's windows reflect the passing landscape of lush green fields and distant mountains, creating a mesmerizing blend of nature and technology.

**40.** A sleek, silver airplane glides gracefully through a clear blue sky, its wings cutting through the air with precision. As it descends, the sun glints off its polished surface, casting a radiant glow.

**41.** A spirited individual rides a vintage bicycle along a sunlit, tree-lined path, wearing a casual outfit of a white t-shirt, denim shorts, and sneakers. The scene captures the golden hour, with sunlight.

**42.** A young man with long, flowing hair sits on a rustic wooden stool in a cozy, dimly lit room, strumming an acoustic guitar. He wears a vintage denim jacket over a white t-shirt and faded jeans, his fingers skillfully moving across the strings.

**43.** A young person, dressed in a vibrant red jacket and black jeans, rides a sleek electric scooter through a bustling city street. The scene captures the energy of urban life, with towering skyscrapers and colorful storefronts lining the background.

**44.** A young person, wearing a cozy gray hoodie and black-rimmed glasses, sits in a dimly lit room, intensely focused on a video game. The glow from the TV screen illuminates their face, highlighting their concentration.

**45.** A young woman with glasses is jogging in the park wearing a pink headband.

| Prompt ID | Aesthetic Quality | | | | | Motion Smoothness | | | | | Semantic Coherence | | | | |
|---|---|---|---|---|---|---|---|---|---|---|---|---|---|---|---|
| | A | B | C | D | E | A | B | C | D | E | A | B | C | D | E |
| 1 | 0.966667 | 0.9 | 1.433333 | 2.066667 | 2.133333 | 1.966667 | 1.233333 | 0.9 | 1.3 | 2.1 | 1.533333 | 1.3 | 0.833333 | 1.7 | 2.133333 |
| 2 | 0.966667 | 0.833333 | 1.366667 | 2.2 | 2.133333 | 1.833333 | 1.1 | 0.966667 | 1.666667 | 1.933333 | 1.633333 | 0.9 | 0.966667 | 2.066667 | 1.933333 |
| 3 | 1.1 | 1.033333 | 1.633333 | 2.066667 | 1.666667 | 1.833333 | 1.133333 | 1.033333 | 1.3 | 2.2 | 1.433333 | 1.1 | 0.966667 | 1.8 | 2.2 |
| 4 | 1.166667 | 0.733333 | 1.433333 | 2.3 | 1.866667 | 1.7 | 1.1 | 1.033333 | 1.6 | 2.066667 | 1.366667 | 1.3 | 1.033333 | 1.633333 | 2.166667 |
| 5 | 0.933333 | 1.166667 | 1.666667 | 2.166667 | 1.566667 | 1.733333 | 1.133333 | 0.8 | 1.5 | 2.333333 | 1.433333 | 1.433333 | 0.966667 | 1.8 | 1.866667 |
| 6 | 1.1 | 0.8 | 1.433333 | 2.066667 | 2.1 | 2 | 1.166667 | 0.866667 | 1.433333 | 2.033333 | 1.533333 | 1.1 | 0.8 | 2 | 2.066667 |
| 7 | 1.4 | 0.866667 | 1.366667 | 2.033333 | 1.833333 | 1.866667 | 1.133333 | 0.866667 | 1.4 | 2.233333 | 1.533333 | 1.1 | 0.833333 | 2 | 2.033333 |
| 8 | 1 | 0.833333 | 1.633333 | 2.1 | 1.933333 | 1.833333 | 1.266667 | 0.866667 | 1.5 | 2.033333 | 1.433333 | 1.166667 | 0.933333 | 1.766667 | 2.2 |
| 9 | 1.066667 | 0.733333 | 1.533333 | 2.166667 | 2 | 1.833333 | 1.3 | 0.933333 | 1.166667 | 2.266667 | 1.766667 | 1.066667 | 0.866667 | 1.9 | 1.9 |
| 10 | 1.266667 | 0.866667 | 1.366667 | 2.1 | 1.9 | 1.7 | 1 | 0.933333 | 1.666667 | 2.2 | 1.466667 | 1.133333 | 0.9 | 1.733333 | 2.266667 |
| 11 | 1 | 0.866667 | 1.533333 | 2.166667 | 1.933333 | 2.033333 | 1.233333 | 0.933333 | 1.4 | 1.9 | 1.5 | 0.866667 | 0.866667 | 1.933333 | 2.333333 |
| 12 | 1 | 1 | 1.433333 | 2.133333 | 1.933333 | 1.866667 | 1.3 | 0.833333 | 1.4 | 2.1 | 1.433333 | 1.266667 | 0.933333 | 1.766667 | 2.1 |
| 13 | 1.2 | 1 | 1.433333 | 2.1 | 1.766667 | 1.933333 | 1.266667 | 0.9 | 1.333333 | 2.066667 | 1.633333 | 1 | 0.866667 | 1.966667 | 2.033333 |
| 14 | 1.133333 | 1.033333 | 1.6 | 1.966667 | 1.766667 | 1.733333 | 1.233333 | 1.233333 | 1.166667 | 2.133333 | 1.3 | 1.2 | 0.866667 | 2.133333 | 2 |
| 15 | 1.333333 | 0.833333 | 1.266667 | 2.233333 | 1.833333 | 2.066667 | 1.133333 | 0.533333 | 1.766667 | 2 | 1.733333 | 1.1 | 0.8 | 1.866667 | 2 |
| 16 | 0.933333 | 0.866667 | 1.666667 | 2.266667 | 1.766667 | 1.933333 | 0.933333 | 0.966667 | 1.6 | 2.066667 | 1.433333 | 1.066667 | 0.966667 | 1.9 | 2.133333 |
| 17 | 1.366667 | 1.2 | 1.333333 | 2 | 1.6 | 1.833333 | 1.133333 | 0.866667 | 1.433333 | 2.233333 | 1.2 | 1.266667 | 0.966667 | 2.1 | 1.966667 |
| 18 | 0.966667 | 1 | 1.566667 | 2.233333 | 1.733333 | 2.033333 | 1.2 | 1 | 1.333333 | 1.933333 | 1.633333 | 1.2 | 0.8 | 1.8 | 2.066667 |
| 19 | 1.4 | 0.8 | 1.433333 | 2.233333 | 1.633333 | 1.933333 | 1.233333 | 0.766667 | 1.433333 | 2.133333 | 1.7 | 1.2 | 0.7 | 1.9 | 2 |
| 20 | 1.166667 | 0.766667 | 1.6 | 1.966667 | 2 | 1.7 | 1.1 | 1.033333 | 1.533333 | 2.133333 | 1.5 | 1.3 | 0.733333 | 1.666667 | 2.3 |
| 21 | 1.233333 | 0.7 | 1.5 | 2.066667 | 2 | 1.833333 | 1.366667 | 0.766667 | 1.5 | 2.033333 | 1.5 | 1.1 | 0.933333 | 1.833333 | 2.133333 |
| 22 | 1.066667 | 0.933333 | 1.466667 | 2.1 | 1.933333 | 2 | 1.166667 | 0.833333 | 1.466667 | 2.033333 | 1.5 | 1.3 | 0.7 | 1.766667 | 2.233333 |
| 23 | 1.166667 | 1.033333 | 1.466667 | 2.133333 | 1.7 | 1.866667 | 1.066667 | 0.7 | 1.6 | 2.266667 | 1.666667 | 0.866667 | 0.833333 | 1.966667 | 2.166667 |
| 24 | 1.066667 | 0.833333 | 1.433333 | 2.3 | 1.866667 | 1.933333 | 0.933333 | 0.933333 | 1.5 | 2.166667 | 1.5 | 1.366667 | 0.9 | 1.8 | 1.933333 |
| 25 | 1.3 | 0.733333 | 1.533333 | 2.2 | 1.733333 | 1.8 | 1.033333 | 0.9 | 1.6 | 2.166667 | 1.5 | 1.066667 | 0.966667 | 1.666667 | 2.3 |
| 26 | 1.2 | 0.6 | 1.7 | 2 | 2 | 1.7 | 1.366667 | 1 | 1.6 | 1.833333 | 1.633333 | 1.233333 | 0.833333 | 1.766667 | 2.033333 |
| 27 | 1.033333 | 1 | 1.666667 | 2.066667 | 1.733333 | 1.8 | 1.466667 | 1 | 1.3 | 1.933333 | 1.533333 | 1.166667 | 0.766667 | 1.9 | 2.133333 |
| 28 | 1.033333 | 0.866667 | 1.6 | 2.033333 | 1.966667 | 1.7 | 1.2 | 0.9 | 1.6 | 2.1 | 1.4 | 1.1 | 1.1 | 1.933333 | 1.966667 |
| 29 | 1.033333 | 0.833333 | 1.5 | 2.166667 | 1.966667 | 1.933333 | 1.166667 | 1.033333 | 1.366667 | 2 | 1.466667 | 1.2 | 0.766667 | 1.966667 | 2.1 |
| 30 | 1.2 | 0.766667 | 1.666667 | 1.833333 | 2.033333 | 1.7 | 1.2 | 0.933333 | 1.466667 | 2.3 | 1.633333 | 1.2 | 0.633333 | 1.833333 | 2.2 |
| 31 | 1.133333 | 0.833333 | 1.5 | 1.966667 | 2.066667 | 1.666667 | 1.266667 | 0.766667 | 1.6 | 2.2 | 1.533333 | 1.066667 | 0.933333 | 1.933333 | 2.033333 |
| 32 | 1.266667 | 0.766667 | 1.433333 | 1.933333 | 2.1 | 1.9 | 1.233333 | 0.733333 | 1.6 | 2.033333 | 1.6 | 1.066667 | 0.8 | 1.866667 | 2.166667 |
| 33 | 1.166667 | 0.9 | 1.5 | 2.166667 | 1.766667 | 1.733333 | 1.333333 | 0.833333 | 1.533333 | 2.066667 | 1.366667 | 1.333333 | 0.833333 | 1.9 | 2.066667 |
| 34 | 1 | 0.866667 | 1.533333 | 2.066667 | 2.033333 | 1.966667 | 0.866667 | 1 | 1.5 | 2.166667 | 1.4 | 1.133333 | 0.8 | 2.1 | 2.066667 |
| 35 | 1.066667 | 0.866667 | 1.5 | 2.233333 | 1.833333 | 2.1 | 0.9 | 0.933333 | 1.433333 | 2.133333 | 1.6 | 1.566667 | 0.7 | 1.433333 | 2.2 |
| 36 | 1.2 | 1 | 1.466667 | 2.066667 | 1.733333 | 1.966667 | 1.166667 | 0.9 | 1.433333 | 2.033333 | 1.266667 | 1.2 | 0.766667 | 2.166667 | 2.1 |
| 37 | 1.033333 | 1.033333 | 1.5 | 2.2 | 1.733333 | 2 | 1.066667 | 1 | 1.3 | 2.133333 | 1.466667 | 1.133333 | 0.866667 | 2.033333 | 2 |
| 38 | 1 | 0.966667 | 1.433333 | 1.9 | 2.2 | 1.866667 | 1.433333 | 0.933333 | 1.433333 | 1.833333 | 1.7 | 1.1 | 1.033333 | 1.733333 | 1.933333 |
| 39 | 1.033333 | 0.733333 | 1.566667 | 2.1 | 2.066667 | 1.866667 | 1.166667 | 0.866667 | 1.433333 | 2.166667 | 1.766667 | 1.166667 | 0.7 | 1.666667 | 2.2 |
| 40 | 0.933333 | 1.033333 | 1.433333 | 2.166667 | 1.933333 | 1.833333 | 1.1 | 0.9 | 1.466667 | 2.2 | 1.366667 | 1.4 | 0.833333 | 2 | 1.9 |
| 41 | 1.066667 | 0.866667 | 1.733333 | 2.033333 | 1.8 | 1.833333 | 1.066667 | 0.866667 | 1.466667 | 2.266667 | 1.5 | 1.166667 | 1 | 1.766667 | 2.066667 |
| 42 | 1.033333 | 0.966667 | 1.566667 | 2.133333 | 1.8 | 1.933333 | 1.166667 | 0.833333 | 1.6 | 1.966667 | 1.6 | 1.233333 | 1 | 1.833333 | 1.833333 |
| 43 | 1 | 0.933333 | 1.9 | 1.9 | 1.766667 | 1.866667 | 1.166667 | 0.8 | 1.466667 | 2.2 | 1.5 | 1.233333 | 0.966667 | 1.666667 | 2.133333 |
| 44 | 1 | 0.866667 | 1.333333 | 2.366667 | 1.933333 | 1.866667 | 1.1 | 0.866667 | 1.633333 | 2.033333 | 1.4 | 1.266667 | 0.766667 | 2.033333 | 2.033333 |
| 45 | 0.933333 | 0.866667 | 1.6 | 2.233333 | 1.866667 | 1.933333 | 1.166667 | 0.9 | 1.533333 | 1.966667 | 1.566667 | 1.133333 | 0.933333 | 1.733333 | 2.133333 |
| 46 | 1.133333 | 0.7 | 1.433333 | 2.233333 | 2 | 1.766667 | 1.166667 | 1.066667 | 1.166667 | 2.333333 | 1.4 | 1.133333 | 0.9 | 2.1 | 1.966667 |
| 47 | 1.133333 | 0.766667 | 1.6 | 2.2 | 1.8 | 1.966667 | 1.033333 | 0.933333 | 1.533333 | 2.033333 | 1.366667 | 1.033333 | 1.166667 | 1.833333 | 2.1 |
| 48 | 1.166667 | 0.933333 | 1.6 | 2.066667 | 1.733333 | 1.666667 | 1.233333 | 0.733333 | 1.666667 | 2.2 | 1.833333 | 1.066667 | 0.866667 | 1.733333 | 2 |
| 49 | 1.133333 | 1.066667 | 1.3 | 2.133333 | 1.866667 | 2.1 | 1.133333 | 0.833333 | 1.466667 | 1.966667 | 1.566667 | 1.333333 | 0.733333 | 1.7 | 2.166667 |
| 50 | 1.2 | 0.766667 | 1.566667 | 2.133333 | 1.833333 | 1.966667 | 0.866667 | 0.833333 | 1.566667 | 2.266667 | 1.733333 | 0.966667 | 0.966667 | 1.866667 | 1.966667 |

Figure 7: Detail results of user study (A: PyramidFlow, B: ours, C: HuanyuanVideo, D: CogVideoX, E: T2V-Turbo)

**46.** A young woman with long, dark hair sits alone in a dimly lit room, her face illuminated by the soft glow of a nearby lamp. Tears stream down her cheeks, glistening in the light, as she clutches a crumpled letter in her trembling hands.

**47.** A young woman with long, flowing hair sits at a grand piano in a dimly lit room, her fingers gracefully dancing across the keys. She wears a flowing white dress that contrasts beautifully with the dark wood of the piano.

**48.** a child is playing the guitar in a flower garden.

**49.** a couple of friends is biking in a living room.

**50.** a group of school children is seen walking together, with smartphones.

## C   Ablation Study Settings

### C.1   Evaluation Protocol

To address the computational challenges of comprehensive evaluation, we employ a *customized VBench* as our primary assessment methodology. This choice is motivated by the fact that a full VBench evaluation requires over 160 hours per model variant on a standard NVIDIA A100 GPU, rendering full metric computation impractical for iterative ablation studies. Our customized protocol uses a carefully selected subset of 50 video-generation prompts. All ablation experiments strictly adhere to this fixed prompt set, ensuring direct comparability across architectural variants while reducing the average evaluation time to 4 hours per model.

For these ablation studies, we report seven metrics that can be computed with this prompt subset: two visual consistency metrics—*subject consistency*, *background consistency*; two motion-related metrics—*temporal flickering*, *motion smoothness*; two visual quality-related metrics—*aesthetic quality*, *image quality*; and one video-text alignment metric—*overall consistency*. In Table 4, the Avg. Score is the arithmetic mean of these metrics.

### C.2   Experimental Design

Our ablation study of architecture design adopts a strategic weight initialization approach to enable efficient hypothesis testing. We first pre-train the model with attention operation and then initialize part of the Mamba layer's projection matrices the using pre-trained weights, following the technique in (Wang et al., 2024c). Subsequent training is constrained to 20,000 iterations with a learning rate 1e-4, using the same training dataset for all models. This design ensures that each architectural variant undergoes identical optimization conditions, with only the target module parameters being modified between experimental conditions.

## D   Training Details.

### D.1   Multi-stage Training

To efficiently pre-train the M4V model, we adopt a progressive training strategy. The process begins with text-to-image (T2I) training at 384p resolution. During the subsequent text-to-video (T2V) pre-training phase, we gradually increase the resolution from 384p to 768p and extend the video length from 57 to 121 and 241 frames, training with a combination of image and video data. This staged approach ensures stable adaptation and longer token sequences. For the T2I phase, we follow the training settings from (Jin et al., 2024), using our own image dataset.

**T2V Training.** Direct training on 5-second (121-frame) videos led to very slow convergence. To address this, we first trained on 2-second (57-frame) video data at 384p resolution. This stage utilized the WebVid10M, OpenSora-Plan 1M, and OpenVid1M datasets. The 2-second T2V training was conducted using a learning rate of $1 \times 10^{-4}$, 64 GPUs, and 40k steps. We then transitioned to training on 5-second (121-frame) videos using the same datasets but with extended frame lengths. This phase used the same learning rate, 64 GPUs, and 60k steps.

**Upscaling to 768p.** To further improve visual fidelity and motion smoothness, we performed training on 768p videos with lengths of 121 or 241 frames. This step significantly enhanced video clarity and extended generation duration. At this stage, only the OpenSora-Plan 1M and OpenVid1M datasets were used, due to the relatively lower quality of WebVid10M. The model was trained with a learning rate of $5 \times 10^{-5}$, using 128 GPUs for 20k steps.

**Quality Tuning with Synthetic Data.** We observed that existing datasets lacked sufficient motion diversity (e.g., walking, running), limiting generalization in dynamic scenarios. To alleviate this, we synthesized approximately 80,000 videos using models such as HunyuanVideo. These were generated from GPT-4o prompts focused on various subject motions. In the final training stage, we incorporated these generated videos alongside OpenSora-Plan 1M and OpenVid1M, training the model with a learning rate of $5 \times 10^{-5}$, using 128 GPUs for 30k steps.

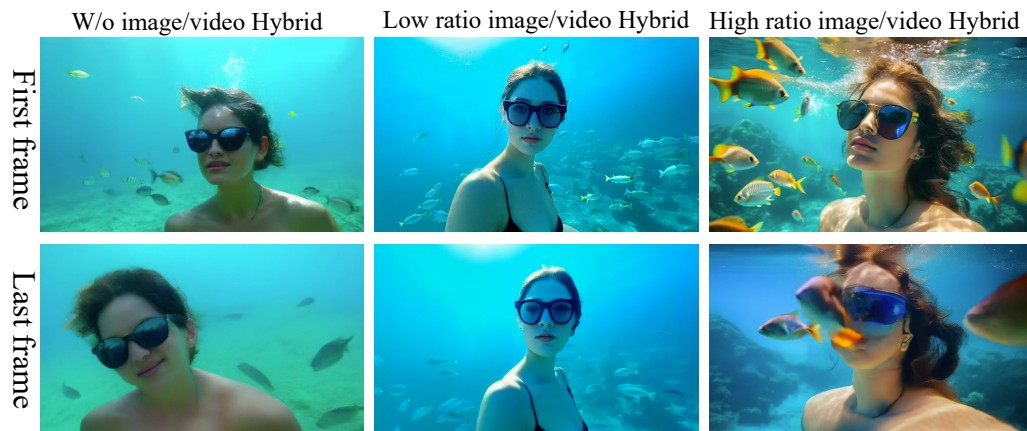

| W/o image/video Hybrid | Low ratio image/video Hybrid | High ratio image/video Hybrid |

Figure 8: Ablation study of different image/video ratio.

**Reward Learning.** To further enhance aesthetic quality in later frames, we introduced a post-training phase after the main training. This phase employed two reward models: one for aesthetic scoring (Wu et al., 2023; Liao et al., 2025) and another for text-image alignment (Radford et al., 2021). We set the reward loss weight to 0.1 and sampled the final 8 latent frames for fine-tuning. This phase used a learning rate of $1 \times 10^{-5}$, 64 GPUs, and ran for 10k steps.

**Effect of Different Image/Video Ratio.** In our experiments, we observe that a higher image-to-video ratio in the training data often leads to subject deformation in later frames. This may be due to the training gradients from image supervision dominating the learning process, causing the autoregressive model to overemphasize the generation of the first frame. Therefore, selecting an appropriate image-to-video mixing ratio is critical for improving the performance of the autoregressive model, as illustrated in the left part of Figure 8. We adopt an image:video ratio of 1:8 as our default setting.

### D.2 PRE-TRAINING INITIALIZATION

To accelerate the learning process of the Mamba-based model, we first pre-train the model using attention operations, at the resolution of 384p. Following the strategy proposed in Mamba-in-LLaMA (Wang et al., 2024c), we initialize the Mamba layers using the pre-trained attention weights. Specifically, we initialize the projection matrices $B$, $C$, and the input projection $x$ in the Mamba block with the weights $W_q$, $W_k$, and $W_v$ from the attention layer, respectively.

After this initialization, we perform additional fine-tuning to adapt the remaining uninitialized weights, such as $A$ and $\Delta$. This fine-tuning stage is conducted with a learning rate of $1 \times 10^{-4}$, using 64 GPUs for 20k steps at the resolution of 384p.

## E ADDITIONAL VISUAL RESULTS.

### E.1 TEXT-TO-VIDEO RESULTS.

We show more visualisation results in Fig. 9. Our advanced design allows the model to create videos that are visually consistent and aesthetically high-quality.

### E.2 IMAGE-TO-VIDEO RESULTS.

Since our model is an autoregressive diffusion model, it is inherently suited for the task of generating videos from images. Specifically, by setting a given image as the initial frame, the model can autoregressively generate the subsequent frames. We show some results in Fig. 10.

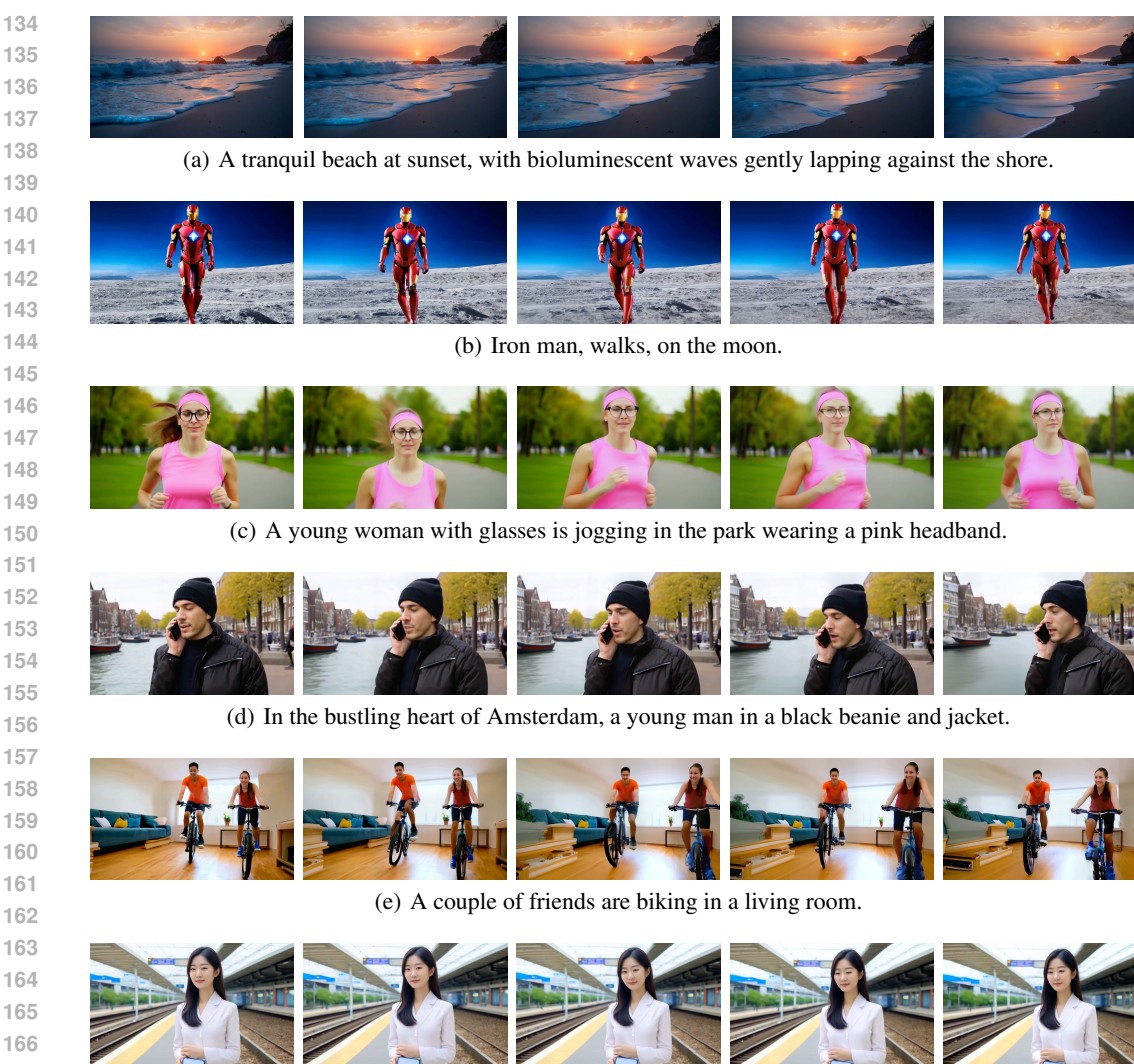

(a) A tranquil beach at sunset, with bioluminescent waves gently lapping against the shore.

(b) Iron man, walks, on the moon.

(c) A young woman with glasses is jogging in the park wearing a pink headband.

(d) In the bustling heart of Amsterdam, a young man in a black beanie and jacket.

(e) A couple of friends are biking in a living room.

(f) A young Japanese woman standing waiting for a train outside station.

Figure 9: Visualization of text-to-video generation results which are generated at 5s, 768p, 24fps.

### E.3 ADDITIONAL IMPROVEMENTS

In video generation, while the image quality, aesthetics, and motion of the earlier frames are generally good, the image quality of later frames tends to degrade. This is primarily due to the accumulation of errors from the previously generated frames, which impacts the subsequent ones. To prevent this error propagation, we propose the use of pure T2I-based correction. Specifically, when generating the first frame, the model effectively operates in an unconditional mode, similar to the unconditioned version of text generation. This allows us to leverage the model's strong T2I capabilities to guide the autoregressive generation of subsequent frames.

Our approach introduces a novel strategy for adjusting the flow prediction during the inference stage. Initially, at the final stage of the pyramid, the model predicts a low-quality, aesthetically suboptimal flow velocity $v_l$. We then compute the predicted $x_0$, and using forward noise addition, we re-input it back into the model to correct the quality. This process ensures that the autoregressive model's limitations in fitting high-quality video frames are mitigated by leveraging the model's capability in fitting high-quality image data during T2I tasks.

The self-quality guidance formula is defined as:

$$v_l = M(x_t, f, \varnothing) + w_p \cdot (M(x_t, f, p) - M(x_t, f, \varnothing)) \tag{8}$$

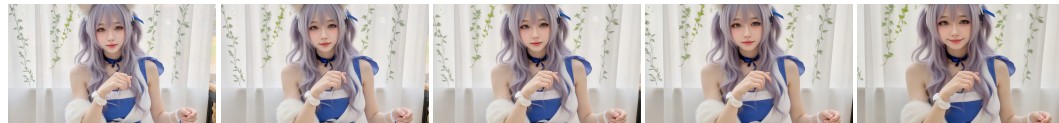

(a) A person riding a bicycle on a wet road. The cyclist is wearing a white blouse with a black tie, a black skirt, black tights, and black shoes.

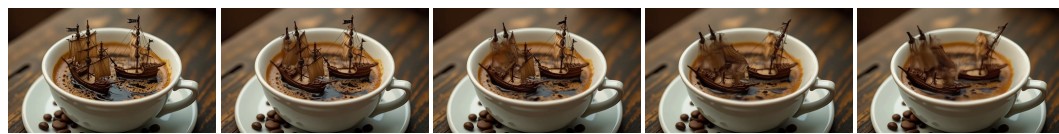

(b) A person sitting on the floor with their legs crossed. The individual has long, wavy hair in shades of purple and white.

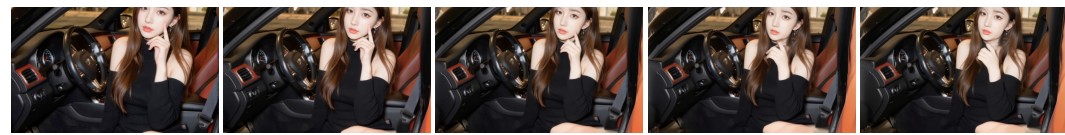

(c) A photorealistic close-up video of two pirate ships battling each other as they sail inside a cup of coffee.

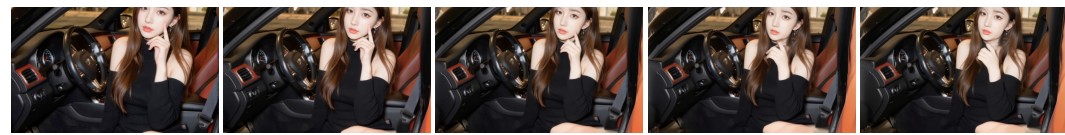

(d) A woman sitting in the driver's seat of a car. The woman has long dark hair and is wearing a black sleeveless top and orange tights.

Figure 10: Visualization of image-to-video generation results which are generated at 5s, 768p, 24fps.

Table 7: Effect of self-guidance on *customized VBench* prompts.

|  | Sub-Cons | BG-Cons | Temp-Flick | Motion-Smooth | Aes-Qual | Img-Qual | Overall-Cons |
|---|---|---|---|---|---|---|---|
| w/o self-guidance | 95.66 | 96.11 | 98.62 | 99.38 | 63.61 | 64.69 | 23.89 |
| w/ self-guidance | 95.60 | 96.12 | 98.61 | 99.38 | 63.89 | 66.38 | 26.62 |

$$v_h = M(x_t, \varnothing, \varnothing) + w_p \cdot (M(x_t, \varnothing, p) - M(x_t, \varnothing, \varnothing)) \tag{9}$$

$$v_{\text{sqg}} = v_l + w_{\text{sqg}} \cdot (v_h - v_l), \tag{10}$$

where $M$ represents our model, $f$ denotes the condition frames, and $p$ refers to the text prompt. Since the earlier frames are typically generated with higher quality, we only apply self-quality guidance starting from the 8th latent frame. Additionally, due to the early stages of the denoising pyramid not yet forming the overall structure and content of the image frames, we begin using self-quality guidance only at the later stages of the pyramid when the texture and content become clearer. This functionality is similar to conventional classifier-free guidance, and its hyperparameters can be adjusted during inference depending on the case. Note that for **all** results in our main paper, we do **not** use the self-quality guidance. We consider this additional improvement as an optional but useful plug-in, which users may choose to enable. Table 7 shows the effect of the self-guidance on our customized VBench, with our final-stage model.

### E.4 THE USE OF LARGE LANGUAGE MODELS (LLMS)

We utilize large language models only for grammar checking, style refinement, and language polishing. No LLMs are used for direct content generation or for research ideation.

