# OpenReview forum: "M4V: Multimodal Mamba for Efficient Text-to-Video Generation"
_ICLR.cc/2026/Conference — ICLR 2026 Conference Withdrawn Submission_

### Official Review · Reviewer_Qio2 · 2025-10-21

**Soundness:** 4
**Presentation:** 4
**Contribution:** 3
**Rating:** 6
**Confidence:** 4

**Summary:**

This paper proposed a post-training architecutral modification, which leverages multimodal Mamba, to reduce the computational cost of T2V model while preserving high-quality generation capability.  Extensive experiments demonstrate the effectiveness of M4V model across multiple evaluation metrics.

**Strengths:**

1. The writting is satisfactory with clear motivation and readiblity, well-organized sections and insightful figures, which helps understanding the proposed method.

2. The application of Mamba to reduce and computation cost is reasonable. The authors successfully address the challenges of extending Mamba to multimodal information processing, while achieving a good trade-off between cost and performance, as shown in Table 4.

3. The experiments is comprehensive to demonstrate the overall performance and effect detailed designs.

**Weaknesses:**

1. It is reasonable that applying Mamba would degrade the performance, as it is sub-optimal to the Transformer block. Though the M4V (Pyramidflow) behaves worse than original Pyramidflow, the M4V (Wan2.1) achieves significan improvement over orignal Wan2.1. Is this phenomenon coming from the initialization strategy? In addition, L288 says that M4V (Pyramidflow) is partly initialized from pretrained attention weights, while Table 1 does not show this detail.

2. Further, in Table 1, the best Semantic Score is Wan2.1 rather than M4V. Please correct this.

3. There lacks a specific model variant for user study. Is it built on Pyramidflow or Wan2.1 framework? From the user study in Figure 3, I notice that the M4V behaves worse than HunyuanVideo in Motion Smoothness and Semantic Coherence, while Table 1 demonstrates a contrast results. Could you provide a further explanation?

4. Table 3 supports the effectiveness of text token re-composition in improving video-text consistentcy. It there any other method to help integrate multimodal information? For example, if we ignore the computation cost, how about interleaving the text tokens with video tokens? Such as "[Text];[Video Scale 1];[Text];[Video Scale 2];[Text]". Would this style of integration get better consistency?

5. The authors list 8 types of Mamba scan paths in Section A.1. However, it is not clear how to arrange different types of scanning in different MM-DiM blocks. Does the order of different types significantly influence the model performance?

Minor Questions (It would be better to explain with Experiment Results)

6. How about training M4V from scratch rather than applying initialization?

7. If we ignored the computational cost, what is the performance upper bound of leveraging pure Mamba for the T2V model?

**Questions:**

My questions are listed in the Weaknesses part. If the authors could address my concerns, I would further raise the final score.

---

### Official Review · Reviewer_W5kk · 2025-10-30

**Soundness:** 2
**Presentation:** 2
**Contribution:** 2
**Rating:** 2
**Confidence:** 4

**Summary:**

This paper proposed a multimodal momba framework for efficient text-to-video generation, in which a multimodal diffusion momba (MM-DiM) block has been designed to boost seamless integration of multimodal information and spatiotemporal modeling. From the presented results, the proposed method achieves better score while requiring lower computation.

**Strengths:**

1、The motivation is clear, i.e., employ Mamba architecture as a more efficient alternative for T2V generation.

2、The evaluation is comprehensive, including both objective indicators and user surveys, as well as time-based testing.

**Weaknesses:**

1、The presented results are somewhat confusing. On one hand, the proposed method reduces computational cost by replacing some MMDiT blocks with the proposed MMDiM. On the other hand, it has only been trained on public datasets. Therefore, it is unclear why the proposed method surpasses MMDiT-based methods—trained on curated indoor datasets—in both visual quality and computational efficiency, especially considering that it is initialized from an image model.

2、For efficient text-to-video (T2V) generation, various active explorations have been conducted, such as model quantization and step distillation, which have demonstrated significant effectiveness. There is insufficient evidence to justify replacing the well-established MMDiT architecture with MMDiM.

3、Based on the presented results, the generated videos exhibit limited motion dynamics and low aesthetic quality.

4、It is recommended to explore the compatibility of this method with model-agnostic acceleration techniques, such as TeaCache (Timestep Embedding Tells: It's Time to Cache for Video Diffusion Models).

**Questions:**

Please refer to the weakness above.

---

### Official Review · Reviewer_5Gre · 2025-10-31

**Soundness:** 2
**Presentation:** 3
**Contribution:** 2
**Rating:** 2
**Confidence:** 4

**Summary:**

This paper proposes M4V, a text-to-video generation framework that replaces Transformer blocks with a novel MultiModal Diffusion Mamba (MM-DiM) block. The core claim is that this adaptation significantly reduces computational complexity while maintaining generation quality. The authors demonstrate results on two base architectures (PyramidFlow and Wan2.1) and employ additional training strategies like reward learning.

**Strengths:**

1. The paper addresses a timely and important problem: reducing the high computational cost of transformer-based video generation models.
2. The proposed MM-DiM block presents a structured approach to adapting the Mamba architecture for multimodal, spatiotemporal data, featuring a detailed token re-composition strategy.
3. The experimental section is extensive, including benchmarks on VBench, human evaluation, and ablation studies.

**Weaknesses:**

1. Insufficient comparison with efficient alternatives: The paper fails to situate its work within the broader landscape of efficiency-focused methods. A critical omission is a comparison with other prevalent techniques for reducing the complexity of attention, such as sparse attention (e.g. SpargeAttention), or other efficient transformers. Without this, it is impossible to judge whether Mamba is the most effective path to efficiency or if similar gains could be achieved with more established, optimized methods.
2. Unclear novelty and missing comparison with prior works: The core technical idea bears significant conceptual similarity to prior work like "Scaling Diffusion Mamba with Bidirectional SSMs for Efficient Image and Video Generation" (Arxiv:2405.15881), which also explores Mamba for diffusion-based generation (including video generation). Thus, the presented paper's novelty remains unclear.
3. Disconnected contributions: The use of a separate reward tuning stage appears to be a method to boost final benchmark scores. However, this technique is orthogonal to the paper's main contribution (the MM-DiM architecture). Its prominent use raises a critical question: is the quality of the pure Mamba-based model insufficient, requiring an auxiliary, generic performance-boosting method to remain competitive? This undermines the claim that the proposed architecture itself maintains high quality, suggesting it may inherently lead to a quality drop that needs to be compensated for.

**Questions:**

1. Given the existence of numerous other methods for achieving sub-quadratic complexity in transformers (e.g., sparse attention), why was the Mamba pathway chosen? Can you provide ablation studies or arguments demonstrating that your MM-DiM block provides a superior efficiency-quality trade-off compared to these established alternatives?
2. The method in Arxiv:2405.15881 also uses Mamba for diffusion modeling. What are the specific, fundamental differences between your MM-DiM block and their approach, particularly regarding multimodal fusion? Clearly, the authors present multi-modal fusion method, which is novel, but the core architecture is still similar to the prior Mamba-based video diffusion model. Please provide a quantitative and qualitative comparison to clarify the novelty of your contribution.
3. The reward learning stage is presented as an "Additional Improvement." If this stage is removed, what is the performance gap between the M4V model and the baselines? Does the core MM-DiM architecture, without post-training reward tuning, cause a significant drop in quality metrics that this separate technique is required to close?

---

### Note · Authors · 2025-11-15

I have read and agree with the venue's withdrawal policy on behalf of myself and my co-authors.